# Identification of Drought-Tolerance Genes in the Germination Stage of Soybean

**DOI:** 10.3390/biology11121812

**Published:** 2022-12-13

**Authors:** Xingzhen Zhao, Zhangxiong Liu, Huihui Li, Yanjun Zhang, Lili Yu, Xusheng Qi, Huawei Gao, Yinghui Li, Lijuan Qiu

**Affiliations:** 1Institute of Crop Sciences, Chinese Academy of Agricultural Sciences, Beijing 100081, China; 2Institute of Crop Sciences, Gansu Academy of Agricultural Sciences, Lanzhou 730070, China

**Keywords:** soybean, germination, drought tolerance, genome-wide association analysis (GWAS)

## Abstract

**Simple Summary:**

Drought stress inhibits seed germination, making it one of the primary environmental factors adversely affecting food security. Soybean is more sensitive to drought than other food crops, especially in germination stage. The purpose of our research is to identify the loci related to drought tolerance in soybean during germination. We used 410 soybean accessions to induce drought at germination stage by PEG, and conducted genome-wide association study. A total of 26 SNPs were found to be related to drought tolerance, and there were nine SNP markers located in or adjacent to (within 500 kb) previously reported drought tolerance QTLs. These SNP led to our identification of 41 candidate genes related to drought tolerance. The current analyses provide information and tools for subsequent studies and breeding programs for improving drought tolerance.

**Abstract:**

Drought stress influences the vigor of plant seeds and inhibits seed germination, making it one of the primary environmental factors adversely affecting food security. The seed germination stage is critical to ensuring the growth and productivity of soybeans in soils prone to drought conditions. We here examined the genetic diversity and drought-tolerance phenotypes of 410 accessions of a germplasm diversity panel for soybean and conducted quantitative genetics analyses to identify loci associated with drought tolerance of seed germination. We uncovered significant differences among the diverse genotypes for four growth indices and five drought-tolerance indices, which revealed abundant variation among genotypes, upon drought stress, and for genotype × treatment effects. We also used 158,327 SNP markers and performed GWAS for the drought-related traits. Our data met the conditions (PCA + K) for using a mixed linear model in TASSEL, and we thus identified 26 SNPs associated with drought tolerance indices for germination stage distributed across 10 chromosomes. Nine SNP sites, including, for example, Gm20_34956219 and Gm20_36902659, were associated with two or more phenotypic indices, and there were nine SNP markers located in or adjacent to (within 500 kb) previously reported drought tolerance QTLs. These SNPs led to our identification of 41 candidate genes related to drought tolerance in the germination stage. The results of our study contribute to a deeper understanding of the genetic mechanisms underlying drought tolerance in soybeans at the germination stage, thereby providing a molecular basis for identifying useful soybean germplasm for breeding new drought-tolerant varieties.

## 1. Introduction

Drought is one of the most impactful environmental factors affecting agricultural production [1,2]. In recent years, global warming and decreased precipitation have caused frequent droughts [3], resulting in substantial 30% reductions in global crop yields [4]. Irrigation is often limited in certain regions and increases production costs [1,5]. Therefore, much current agricultural research focuses on understanding drought tolerance in crops, seeking to accelerate the development of suitably drought-tolerant cultivars [2,6].

Soybean is an important global food source and a valuable economic crop: it is one of the main sources of plant fat and protein for humans, possesses medicinal value, and is widely used as a raw material in industry [7,8]. Of the legumes, soybean is the most sensitive to water [9]; indeed, drought conditions have reduced annual global soybean production by approximately 40% [10,11]. Seed germination is an extremely important growth stage for plants, and drought stress during germination can reduce the overall number of seedlings by 20% and, in serious cases, can reduce yields by more than 50% [4,12]. This makes the identification of drought-tolerant germplasm and the cultivation of drought-resistant cultivars particularly important in soybean.

Extensive research has been conducted to study drought tolerance in soybean. The main drought tolerance traits identified to date include germination rate (GR) [2,13,14], germination energy (GE) [2], drought index [15], leaf wilting [1,16,17], water use efficiency [18,19], canopy wilt [7,20], fibrous roots [20], and yield under water deficit [21]. Research investigating drought tolerance at the germination stage of soybean has been mainly based on PEG treatment to simulate drought conditions. Vijay et al. [2] employed different concentrations of PEG-6000 to study the drought tolerance of soybean accessions, demonstrating that GR, vigor index, and stress tolerance index can be used to evaluate the drought tolerance of soybeans at the germination stage. Dantas et al. [22] found that the optimum −0.2 MPa is an osmotic potential.

Drought tolerance in plants is a quantitative trait controlled by multiple genes [23]. QTLs (quantitative trait loci) related to drought tolerance have been identified and can be used as molecular markers [24]. There are currently 120 drought-tolerant QTLs represented in SoyBase (Available online: www.soybase.org (accessed on 22 May 2020)); these are primarily distributed on chromosomes 2, 5, 17, and 19 [7,15,20]. Specht et al. [25] constructed a recommended inbred line (RIL) population with 236 lines from a Minsoy × Noir1 cross, and they performed six different levels of water stress experiments over two years; this work identified three QTL loci related to yield under drought stress. Hwang et al. [7] used five RIL populations to study the QTLs controlling leaf wilting, identifying seven stable QTLs. Kaler et al. [24] used 31,260 SNPs to conduct genome-wide association analysis on 373 soybean accessions and identified 47 SNP markers related to WUE (water use efficiency). Liu et al. [14] used 4616 SNPs to conduct genome-wide association analysis on 259 soybean accessions at the germination stage and identified 15 QTLs related to drought tolerance indices during germination.

It is notable that the results from previous studies often differ, owing in part to their various use of diverse different mapping groups, molecular markers, experimental environments, and calculation methods. There are, therefore, relatively few drought-tolerant QTLs that have been separately verified by separate research groups. Due to drought tolerance localization (which is limited by the number of parents), the identified drought tolerance QTL is easily overlooked [26]. As such, the drought tolerance of soybeans during the germination stage requires further study.

In the present study, we aimed to identify the genetic mechanisms underlying drought tolerance at germination stage using association analysis. Here, we examined a soybean germplasm diversity panel comprising 410 accessions, which we screened with drought conditions using PEG-6000. We used the data from the germination stage growth assays to calculate the relative germination rate (RGR), relative germination energy (RGE), germination drought tolerance index (GDTI), germination stress index (GSI), and membership function value (MFV) for each of the soybean accessions. We genotyped the 410 accessions using 158,327 SNPs, and then we conducted a whole-genome association analysis. Our GWAS identified 26 SNPs associated with drought tolerance indices at the germination stage, among which nine were associated with two or more indices. There were nine SNP markers located in or adjacent to previously reported drought tolerance QTLs, and 41 candidate genes related to drought tolerance in the germination stage were identified. Our results will provide a molecular basis for identifying drought-tolerant germplasm and will help develop new soybean cultivars that exhibit strong drought tolerance at the germination stage.

## 2. Materials and Methods

### 2.1. Plant Materials

An amount of 410 soybean accessions were obtained from the Chinese National Soybean Genebank (CNSGB), including 110 non-Chinese accessions (from 8 countries, including the United States and Russia) and 300 Chinese domestic accessions (from 27 Chinese provinces, with many accessions from the Northeastern provinces of Heilongjiang and Liaoning, in the known center of soybean domestication) (Appendix A). Pilot screening identified the set of 6 accessions (Appendix A)—each with different levels of drought tolerance—that we used to initially optimize the PEG-6000 concentration we used for the larger-scale screen of the full 410 accession germplasm diversity panel.

### 2.2. Methods

#### Optimum PEG-6000 Concentration Screening

We used six soybean accessions with different levels of drought tolerance at the germination stage for this drought treatment optimization process. Seeds of uniform size were selected from each accession; these were sterilized with 0.1% HgCl_2_ for 30 s, washed with sterile water 2–3 times, and dried with filter paper. Twenty seeds per genotype were used in each of three replications. The seeds were placed on wetted filter paper in 9-cm-diameter Petri dishes to evaluate the growth performance and phenotypic variation in seedlings. We then added 15 mL of PEG-6000 solution to each dish at the following concentrations: 0% (CK), 10%, 15%, 20%, and 25% (W/W). The culture dish was then placed in an artificial climate incubator at constant temperature and humidity (temperature 25 °C and humidity 60%), and the appropriate PEG-6000 concentrations were added to each of the treatments each two days to keep the germinating bed moist. Germination was assessed at 24-h intervals for 8 consecutive days. The GR, GE, germination index (GI), and germination drought index (GDI) of different PEG-6000 concentrations for each accession were compared to determine the optimal concentration of PEG-6000, according to the method of Ku et al. [27] and Thabet et al. [28].
$$\mathrm{GR}\left(\%\right)=n/N\times 100;$$
$$\mathrm{GE}\left(\%\right)=m/N\times 100;$$
$$\mathrm{GI}=\sum \left(\mathrm{DG}/\mathrm{DT}\right);$$
$$\mathrm{GDI}=\left(1.00\right)\mathrm{nd}2+\left(0.75\right)\mathrm{nd}4+\left(0.50\right)\mathrm{nd}6+\left(0.25\right)\mathrm{nd}8.$$

In these formulas, “n” is the number of germinated seeds on the eighth day, “m” is the number of seeds germinated on the sixth day [27], “N” is the total number of seeds, “DG” is the total number (cumulative) number of seeds germinated on each measuring two days, “DT” is the germination days corresponding to DG, “nd2”, “nd4”, “nd6”, and “nd8” are the germination rates of seeds on the second day, the fourth day, the sixth day, and the eighth day, respectively, and “1.00”, “0.75”, “0.50”, and “0.25” are the drought tolerance coefficients assigned by the corresponding germination days, respectively.

### 2.3. Phenotype Identification and Drought Tolerance Evaluation in the Germination Stage

Using the optimum PEG-6000 concentration (20%), 410 germinating plants from each accession of the panel were tested for traits including GR, GE, GI, and GDI. The RGR, RGE, GDTI, GSI [28], and MFV [27,29] traits were used as evaluation indices to examine the drought tolerance of the materials during the germination stage. The calculation method was as follows.
$${\mathrm{RGR}=\mathrm{GR}}_{T}/{\mathrm{GR}}_{C};$$
$${\mathrm{RGE}=\mathrm{GE}}_{T}/{\mathrm{GE}}_{C};$$
$${\mathrm{GDTI}=\mathrm{GDI}}_{T}/{\mathrm{GDI}}_{C};$$
$${\mathrm{GSI}=\mathrm{GI}}_{T}/{\mathrm{GI}}_{C};$$
$$\mathrm{MFV}=\left(\sum X_{u}\right)/{M,\;X}_{u\;}=(X\;-{\;X}_{\mathrm{Min}})/{(X}_{\mathrm{Max}}-X_{\mathrm{Min}}).$$

In these formulas, “T” is the treatment, “C” is the control (water), “X_u_” is the subordinate function of an indicator of the accessions, “X” is the measured value of an indicator of the accessions, “X_Min_” and “X_Max_” are the minimum and maximum values within the measured value of an indicator of all accessions, and “M” is the number of measured indicators.

### 2.4. Phenotypic Data Analysis

Statistical analysis of all phenotypic data across the four germination-related traits and five drought tolerance indices was conducted using the software SAS PROC GLM. (SAS Institute 1999). The broad-sense heritability (h^2^) [30] of each trait was estimated using the variance components. All of the above variance values can be calculated using the REML method for the SAS VARCOMP procedure.
$$H^{2}=\frac{\sigma_{\mathrm{Geno}}^{2}}{\sigma_{\mathrm{Geno}}^{2}{+\sigma }_{\varepsilon }^{2}}$$

### 2.5. Genotype Identification and Analysis

#### Genotype Identification

Genomic DNA was extracted from soybean seedling leaves according to the methods used by Kisha et al. [31], and DNA quality was detected by 1% agarose gel electrophoresis and a spectrophotometer. A genome-wide genotyping array containing 158,327 SNPs was applied to genotype the 410 accessions using the Illumina Infinium platform, according to the manufacturer’s protocol (Illumina) [32,33]. All SNP genotype data were treated with raw data normalization, clustering, and genotype calling using Illumina Genome Studio Genotyping Module (Illumina). The SNPs with a minor allele frequency (MAF) < 0.05 and missing rates < 0.25 were removed to avoid problems of spurious LD and false positive associations. Finally, 117,811 high-quality SNPs were used for GWAS analysis. The SNPs were distributed relatively evenly across the 20 soybean chromosomes (Appendix A).

### 2.6. Analysis of Gene Diversity, Linkage Disequilibrium, and Population Structure

We used PowerMarker v3.25 software to analyze MAF, PIC, heterozygosity, and gene diversity [34]; PLINK software was used to analyze the attenuation distance of linkage disequilibrium (LD) of the related population [35], and the R language was used for mapping [32]. We used half of the maximum distance for LD attenuation to identify LD blocks; this was the support interval we used for identifying significant SNPs related to a particular trait. We used multivariate analysis to classify the soybean accessions into subgroups, including cluster analysis with a neighbor-joining algorithm, model-based population structure analysis, and principal component analysis (PCA). Cluster analysis and the PCA were performed in TASSEL 5.0. When the eigenvalue is flat, the subgroup structure is determined (after PC4 in our model), and population structural analysis was performed using the admixture program [36].

### 2.7. Genome-Wide Association Analysis

We performed a genome-wide association analysis using a mixed linear model (MLM) that accounted for kinship (K matrix) and population structure (PAC matrix) in TASSEL 5.0 [37]. The Loiselle algorithm [38,39] was used to approximate the kinship coefficient between each pair of accessions in TASSEL 5.0. Significant SNPs were those with −log(*p*) > 4.5 in MLM. Any significant markers positioned within a single LD block were viewed as one QTL region.

## 3. Results

### 3.1. Selection of the Optimal Concentration of PEG-6000

Seeds of the six variously drought-tolerant soybean accessions were treated with five different concentrations of a PEG-6000 solution (Figure 1). As expected, the GR and GE values for all of the accessions decreased as the concentration of PEG-6000 increased. At 25% PEG-6000, the soybeans did not germinate. When treated with 10% PEG-6000 and 15% PEG-6000, there were no significant differences among the genotypes for GR, GE, GI, or GDI compared to the control samples (0% PEG-6000). When treated with 20% PEG-6000, the GR, GE, GI, and GDI of all six accessions were significantly reduced (to varying degrees) compared to the control, and the differences were significant between the 20% PEG-6000 treatment and the control. Therefore, we selected the 20% PEG-6000 concentration for simulating the drought-stress condition for our large-scale, germination-stage screening of the 410 accessions of our soybean germplasm diversity panel.

### 3.2. Phenotype Analysis of Soybean Germplasm at the Germination Stage

#### 3.2.1. Descriptive Analysis of Four Germination-Related Traits and Drought Tolerance Traits

We measured four germination-related traits (GI, GDI, GE, and GR) for germinating seeds of the 410 soybean accessions under 20% PEG-6000 (T) or 0% PEG-6000 (C). Appendix A displays the calculated mean for each trait, the ranges, the standard deviations, and the coefficients of variation. The mean values for the whole drought-treated panel for the GI, GDI, GE, and GR traits were 0.68, 17.49, 13.58%, and 15.07%, respectively, whereas the means for the controls were 7.50, 189.20, 96.33%, and 96.92%. The results of a MIXED model procedure ANOVA (Table 1) identified significant differences among genotypes, treatment, and genotypes × treatment (*p* < 0.001). The treatment mean square was the largest, suggesting that the drought treatment was the most impactful factor.

#### 3.2.2. Analysis of Drought Tolerance

The respective mean values for RGR, RGE, GDTI, GSI, and MFV were 0.16, 0.14, 0.09, 0.08, and 0.15 for the 410 soybean germplasm accessions (Table 2, Figure 2). The coefficient of ranges for RGR, RGE, and MFV were 0–1, the coefficient of variation ranges for GDTI and GSI were 0–0.57 and 0–0.48, respectively, and the maximum coefficient of variation for RGE was 136.51. The minimum coefficient of VC for MFV was 121.81. The results of variance analysis of RGR, RGE, GDTI, and GSI demonstrated that there were significant differences among genotypes for all examined indices (*p* < 0.001), but no significant differences were observed between repeats (Table 3). The results of the correlation analysis demonstrated that there were significant positive correlations for all indices (*p* < 0.001) (Table 4), which could be because the phenotypic values of five drought tolerance indices were calculated on the basis of the germination number. The broad-sense heritability of MFI, GSI, GDTI, RGE, and RGR was high: 90.41%, 87.78%, 88.90%, 90.86%, and 91.43%, respectively, which aid in early selection of offspring.

### 3.3. Analysis of Soybean Genetic Diversity

#### 3.3.1. Analysis of Genetic Diversity and Linkage Disequilibrium

The results of a PowerMarker analysis demonstrated that the mean MAF value for 117,811 SNPs among the 410 accessions of the diversity panel was 0.2228 (ranging from 0 to 0.5030), and the proportion of SNPs with MAF greater than 0.2228 was approximately 46.9%. The mean values of genetic diversity, heterozygosity, and PIC were 0.3043, 0.0237, and 0.2548, respectively, and the ranges were 0–0.5061, 0–0.4070, and 0–0.3843 (Figure 3, Appendix A). The results of a LD analysis demonstrated that the whole-genome mean LD for the diversity panel was r^2^ = 0.3440. The r^2^ value decreased to approximately half of its maximum level once the LD decay distance reached approximately 75 kb (Figure 4). This suggests that LD decayed relatively quickly within the accession of the panel.

#### 3.3.2. Population Genetic Structural Analysis

To avoid false-positive associations due to population stratification, three calculations were executed to study population structure: principal component analysis (PCA), phylogenetic-tree construction, and population-structure analysis with ADMIXTURE. Our PCA analysis based on the SNP data for the whole panel indicated, as expected, that eigenvalues decreased as the number of PCs increased (Figure 5B,C). With fewer than four PCs, the eigenvalues decreased gradually; this suggests that accessions in the diversity panel can be plausibly divided into four subgroups. To better understand the genetic diversity of the soybean germplasm panel, we built a neighbor-joining tree based on the incidence of common alleles between the accessions. This analysis also divided the accessions of the panel into four subgroups (Figure 5A), and the findings are consistent with our results from the PCA.

We again detected a 4-subgroup structure for the accessions of the panel when we conducted a population structure analysis using the ADMIXTURE program (Figure 5D). For the four groups, the G1 accessions were primarily from the United States, Japan, and other countries outside China, the G2 accessions were primarily from Northern China, G3 accessions were from diverse locations within China, and the G4 accessions were primarily from Southern China.

### 3.4. GWAS to Identify SNPs Associated with Drought Tolerance

Our MLM GWAS analysis identified a total of twenty-six SNP loci that were significantly associated with drought-tolerance traits (Table 5; Figure 6); these were distributed on 15 chromosomes. There were 8, 8, 22, 5, and 8 SNPs associated with RGR, RGE, GDRI, GSI, and MFV, respectively. There were five significantly drought-tolerant trait-related SNPs on both chromosomes 1 and 20, four SNPs on chromosome 8, two SNPs each on chromosomes 4, 9, and 15, as well as one significant SNP each for chromosomes 2, 3, 5, 6, 7, 11, 13, 14, and 19. Notably, nine loci were associated with two or more drought-related traits. The amount of phenotypic variation explained by these SNPs was 5.19–9.66%, with an average of 6.99%. It was also highly notable that 21 of the QTLs identified in our GWAS are located within 500 kb of previously identified loci in quantitative genetics analyses of soybean, with nine of these near loci previously associated with drought-related traits. The remaining 16 were for yield-related traits.

## 4. Discussion

Identifying germplasm resources in order to evaluate drought tolerance in soybeans is necessary for drought-tolerant breeding, the study of drought tolerance mechanisms, and the detection of molecular markers [14,24]. Previous results demonstrated that germplasm with high drought tolerance had high rates and uniformity of germination [22]. Germination speed, uniformity, and elongation of young roots were then used to explore the drought tolerance of germplasm, and RGR, RGE, GDTI, and GSI data were used to evaluate drought tolerance [28,55]. In the present study, we first used six accessions with different drought tolerance levels and conducted treatments with different concentrations of PEG-6000. By comparatively analyzing the GR, GE, GI, and GDI, we determined that the optimal concentration of PEG-6000 for a larger scale screen was 20%. We then subjected the 410 accessions of our germplasm diversity panel to drought-stress. Analysis with a linear ANOVA model indicated significant (*p* < 0.001) variation for drought tolerance among genotypes (Table 3). The phenotypic coefficient of CV of traits related to drought tolerance was large (121.8–136.5) (Table 2), suggesting significant phenotypic variation, while drought-related traits displayed highly broad-sense heritability (≥85%).

The effect of drought on crops is multifaceted. The membership function method can be used to synthesize multiple evaluation indices, to avoid the bias of a single index [27,29], and to better evaluate the drought tolerance of soybean [27]. In this study, a total of 26 drought tolerance loci were identified. Among these loci, the number of those detected based on the MFV data was eight; and there were also loci detected based on MFV and the other indices, for example, eight by both MFV and GDRI, seven by both MFV and RGE, six by both MFV and RGR, and three by both MFV and GSI. Of the SNPs associated with the loci of the five drought tolerance indices, nine were associated with two or more traits, among which eight were detected by MFV. As such, MFV is excellent for evaluating drought tolerance, highlighting its utility for evaluating drought tolerance in the germination stage of soybean.

Drought tolerance is a quantitative trait controlled by multiple genes [14,23]. In this study, the distribution of phenotypic index values demonstrated significant variation in the 410 accession panel (Figure 3), and a genome-wide association analysis identified 26 QTLs from the 117,811 SNPs that were related to drought tolerance (Table 5). These results reinforce that drought tolerance during the germination stage is controlled by multiple genes.

Nine of the 26 significant drought tolerance loci were associated with two or more drought-related traits. Two loci (Gm20_34956219 and Gm20_349602658) were associated with five drought tolerance indices; four loci (Gm01_35877607, Gm04_4484515, Gm11_30280479, and Gm20_13921498) were associated with four drought tolerance traits; two loci (Gm09_11414508 and Gm09_18023730) were associated with three drought tolerance traits; and Gm_01_47042336 was associated with two drought tolerance traits. These results are in agreement with previously reported results about the involvement of multiple loci in drought-tolerance responses in soybean [51,56,57]. Moreover, as these drought-tolerance associated loci have been detected several times, our study reinforces that these are relatively stable QTL loci.

We compared the results of the association mapping of drought tolerance with previously studied QTLs within a 500 kb range using Soybase (Available online: http://www.soybase.org (accessed on 22 May 2020)). Of the 26 significant SNPs in this study, nine were located in or near the reported QTLs related to drought tolerance (Table 5). Of these, six were related to canopy wilt [7,20], two were related to drought index [15], two were related to WUE [23,24], and one was related to drought tolerance in the germination stage [49]. Gm19_49449499 is located downstream of the QTL satt513 [13], which is reportedly related to drought tolerance in the germination stage; at the same time, a canopy wilting QTL exists near Gm19_49449499 [7]. Gm20_34956219 was associated with data for five drought tolerance indices in our study; this is located near a wilting canopy QTL [20] and within 14 kb of the WUE marker ss715637488 [24]. Additionally, a QTL locus reported to control root density [54] is located near Gm20_34956219. Wang et al. [6] reported that the more lateral roots that soybean accessions have, the stronger the water absorption and, thus, drought resistance. These are stable QTL loci detected by different research materials and using different research methods, so it is likely that there are drought-tolerant genes within their genomic regions. These drought-tolerance associated markers should be useful for identifying causal genes that can be used to improve drought tolerance in soybeans.

Drought is one of the primary abiotic stresses affecting crop production, and it severely restricts soybean yield [4,12]. There are QTLs related to yield traits near the significant SNPs detected in this study, which are located on chromosomes 1, 4, 5, 6, 8, 9, 11, 15, and 20 (Table 5) [5,40,41,44,45,46,48,49,51,52]. Gm08_4052111 is adjacent to the canopy wilt marker [7], and it is located between regions (ranging from satt390-satt424) related to seed weight [53]. We found that Gm09_11414508 is related to three drought-related traits, and it is positioned within the seed yield marker range satt518-BARC-041991-08155 [52]. Gm08_1438457 is located between Sat_383–BARC-010329-00586, which reportedly controls single seed weight QTL [48]. Genomic regions with multiple associated traits suggest pleiotropy of a single causal gene or the close association of multiple causal genes. Using MAS diagrams, these markers can, in theory, be used for molecular marker-assisted selection to help improve both drought tolerance and yield in soybeans.

Of the SNPs with significant associations detected in this study, and in addition to Gm20_34956219, there are three SNPs located close to reported QTLs associated with root traits. For example, Gm20_36902659 is adjacent to a reported QTL related for lateral root density [43]. Near Gm01_47042336, there are root area locus [43] and root length locus [43]. There are also five drought-associated QTLs that we identified, which have not been reported in previous studies. These are new loci and will, thus, require verification by additional studies.

We used the SoyBase database to identify candidate genes directly associated with the SNPs of our detected QTLs or in nearby genes. We identified 26 candidate regions containing 41 genes. Of these, 12 SNPs related to the drought tolerance indices detected in this study were located within genes, including SNPs causing four non-synonymous mutations, as well as three synonymous mutations; there were also three SNPs in the 3’-UTR of genes and two SNPs positioned in gene introns. Fourteen of the significantly associated SNPs were located in intergenic regions (Appendix A). The results of functionally annotating 41 phytozome genes (Available online: https://phytozome.jgi.doe.gov (accessed on 22 May 2020)) suggested that these candidate genes may have functions related to bidirectional sugar transporter sweet, transferase, exportin, and hydrolase. The genes *Glyma.01g106000* (adjacent site Gm01_35877607) and *Glyma.08g103900* (Gm08_7972856 is located on the gene) regulate root morphology and the expression of a transferase in soybean [58,59], which could be related to drought tolerance in soybean. The gene *Glyma.08g017800* (Gm08_1438457 is located on the gene) improves drought tolerance by regulating the rise and fall of glucose under drought conditions in soybean [6]. Gm01_48619013 is located in an exon of the *Glyma.01g149300* gene (*Glyma 01g35220* in v1.1) (Appendix A), which encodes a methyltransferase PMT21-related protein (Appendix A), while the *Glyma.01g149300* gene improves drought tolerance in soybeans by regulating protein synthesis under drought conditions [60]. The consistency of the associations was tested by comparing the drought tolerance of particular genotypes of Gm01_48619013 (Appendix A) SNP sites, as defined by this study. Drought tolerance in accessions that carry Gm01_48619013-GG genotypes was significantly higher within populations than for genotypes homozygous to alternate alleles.

## 5. Conclusions

In this study, 410 soybean accessions were tested for drought tolerance by simulating drought conditions with 20% PEG-6000. Variance analysis demonstrated that there were significant differences among the genotypes in five drought tolerance indices: RGR, RGE, GDRI, GSI, and MFV. A whole-genome association analysis was performed using 158,327 SNP markers. Twenty-six SNP loci related to drought tolerance during the germination stage were detected. Of these, nine SNP loci were significantly related to two or more drought-tolerance traits, nine loci were near QTL loci reportedly related to drought tolerance, and two SNP-related genes (Glyma.04g055500 and Glyma.13g246400) were associated with drought tolerance in soybeans. It is extremely important to continue studying drought-tolerance genes and markers to assist with the selection and development of drought-tolerant soybean accessions.

## Figures and Tables

**Figure 1 biology-11-01812-f001:**
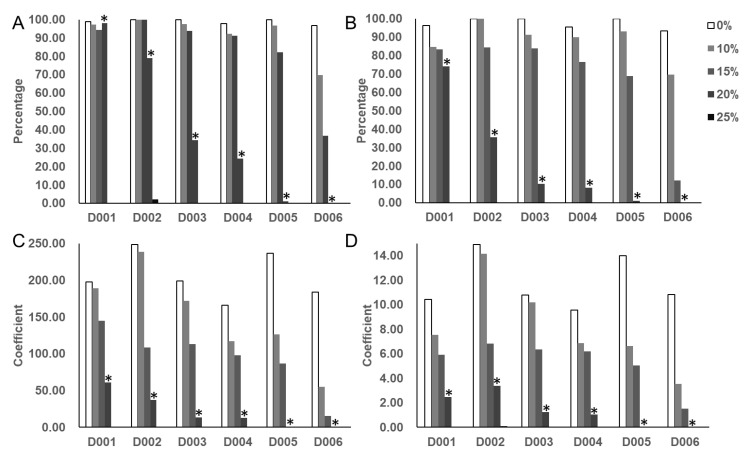
Frequency distribution of four germination-related traits GR (**A**), GE (**B**), GDI (**C**), and GI (**D**) for the six soybean accessions. (GR, germination rate; GE, germination energy; GDI, germination drought index; GI, germination index; 0%, 10%, 15%, 20%, and 25% represent different PEG treatment concentration; *, significantly different from the control. D001-006 represent six soybean accessions (Appendix A)). Each value is the mean of three independent samples.

**Figure 2 biology-11-01812-f002:**
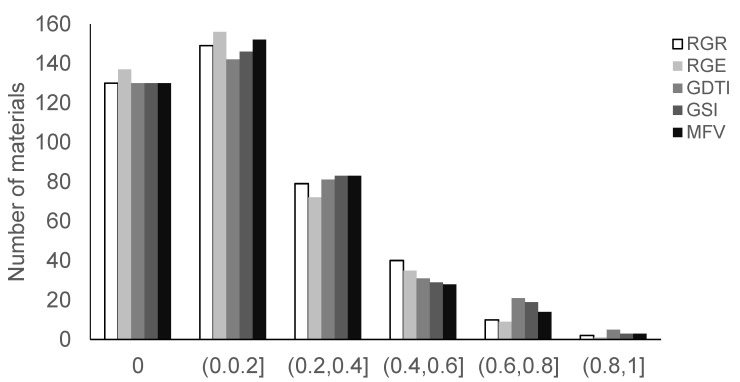
Distribution of five drought tolerance indices (RGR, relative germination rate; RGE, relative germination energy; GDTI, germination drought tolerant index; GSI, germination stress index; MFV, membership function value).

**Figure 3 biology-11-01812-f003:**
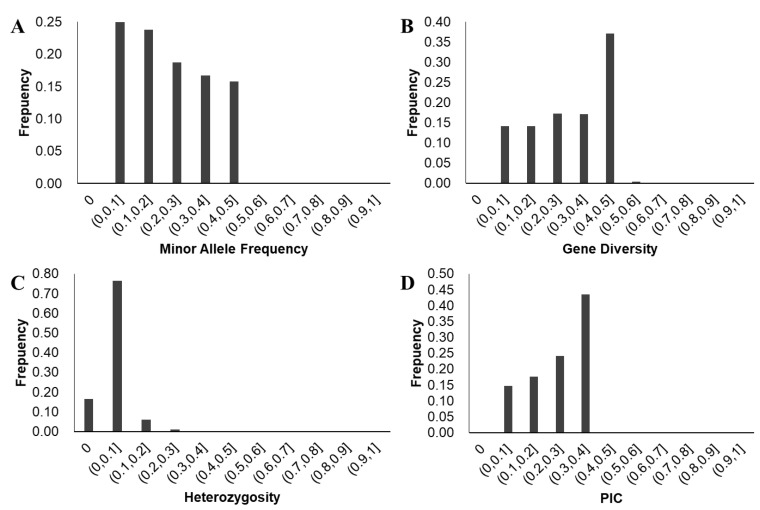
Distribution of the genetic diversity of 136,337 SNPs across 410 accessions. (**A**) is for minor allele frequency; (**B**) is for gene diversity; (**C**) is for heterozygosity; and (**D**) is for polymorphic information content (PIC).

**Figure 4 biology-11-01812-f004:**
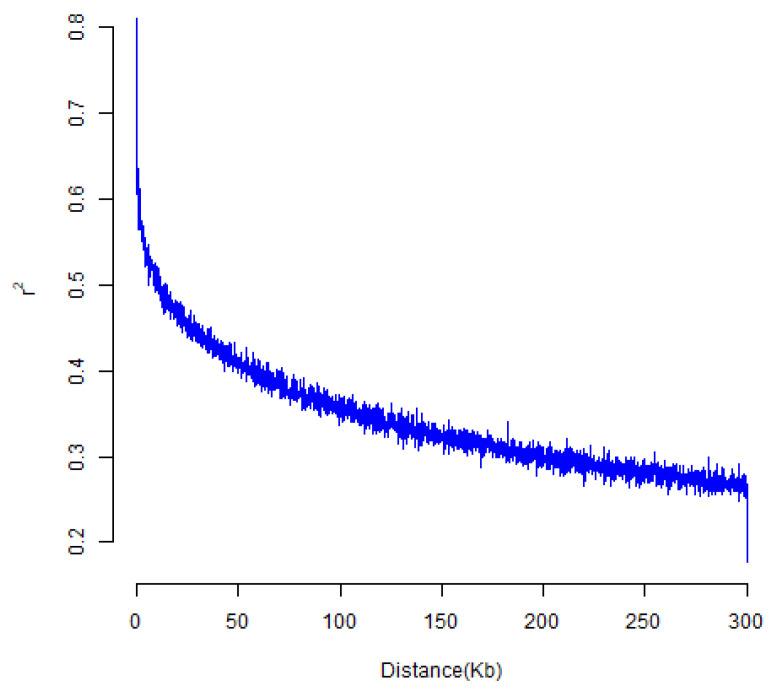
Linkage disequilibrium (LD) decay across soybean genome.

**Figure 5 biology-11-01812-f005:**
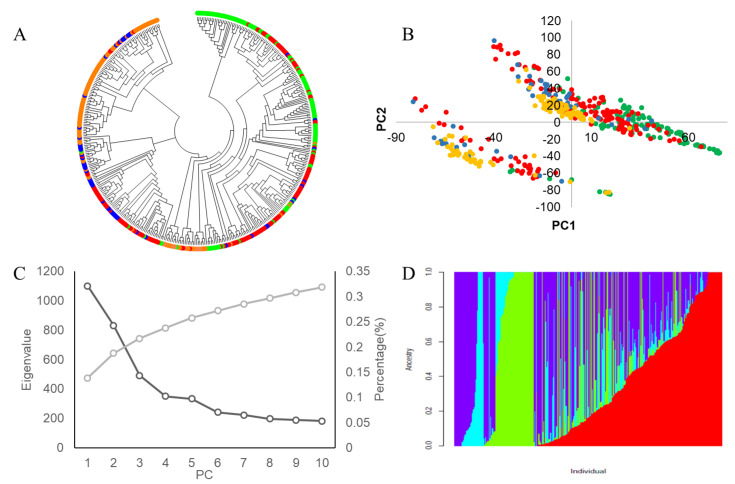
Genetic structure and relatedness of the 410 soybean accessions. (**A**) Neighbor−joining tree constructed using SNP data, foreign soybean germplasm expressed as green; soybean accessions from north are shown in red; those from the Huanghuaihai valley region are shown in blue; and those from the south valley region are shown in orange. (**B**) Principal component analysis for the entire panel of soybean accessions; (**C**) PCA eigenvalue performed by GAPIT using the pruned set of 200K SNP. As presented, the total variance explained by each principal component (PC) decreased from PC1 to PC4 and, after PC4, the variance explained by each further PC remained low and stable; (**D**) Clustering for PCA = 4 for the entire panel of soybean accessions. Each individual is represented by a vertical bar, as well as partitioned into colored segments, with the length of each segment representing the proportion of the individual’s genome from groups when PCA = 4.

**Figure 6 biology-11-01812-f006:**
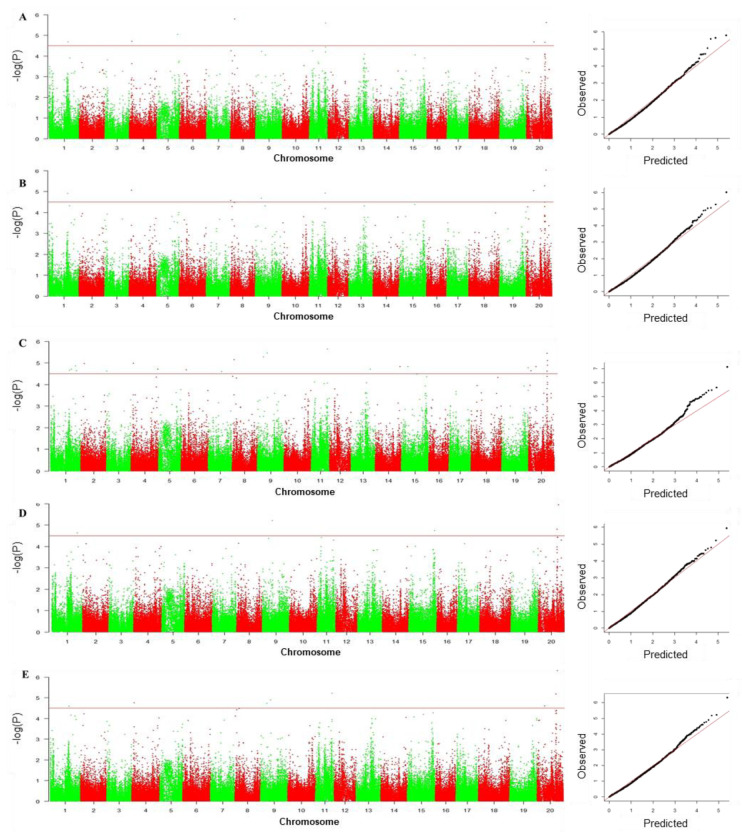
Manhattan and Quantile-quantile plot from the PCA + K model across the five drought tolerance indices RGR (**A**), RGE (**B**), GDTI (**C**), GSI (**D**), and MFV (**E**).

**Table 1 biology-11-01812-t001:** Analysis of variance (ANOVA) of four germination-related traits under 0% PEG and 20% PEG conditions for seeds of the 410 soybean accessions.

Trait	Source	DF	Sum of Square	Mean Square	*F* Value	Pr > F
GR	Geno	409	251,929.70	615.97	7.29	<0.0001
	Treatment	1	437,4687.00	4,374,687.00	51,778.10	<0.0001
	Block/Treat	2	357.14	178.57	2.11	0.1211
	Geno×Treat	409	246,113.60	601.75	7.12	<0.0001
GE	Geno	409	236,174.00	577.44	7.74	<0.0001
	Treatment	1	4,470,355.00	4,470,355.00	59,943.60	<0.0001
	Block/Treat	2	359.92	179.96	2.41	0.0898
	Geno×Treat	409	226,055.60	552.70	7.41	<0.0001
GDI	Geno	409	1,658,797.00	4055.74	10.23	<0.0001
	Treatment	1	19,290,378.00	19,290,378.00	48,679.80	<0.0001
	Block/Treat	2	2222.70	1111.35	2.80	0.0608
	Geno×Treat	409	613,890.90	1500.96	3.79	<0.0001
GI	Geno	409	3440.47	8.41	8.82	<0.0001
	Treatment	1	30,333.58	30,333.58	31,790.80	<0.0001
	Block/Treat	2	9.94	4.97	5.21	0.0055
	Geno×Treat	409	1756.14	4.29	4.50	<0.0001

GR, germination rate; GE, germination energy; GDI, germination drought index; GI, germination index; DF, degree of freedom.

**Table 2 biology-11-01812-t002:** Descriptive statistics of five drought tolerance indices.

Traits	Mean	SD	Skewness	Kurtosis	Range	CV
RGR	0.16	0.20	1.55	2.03	0~1	129.88
RGE	0.14	0.19	1.63	2.40	0~1	136.51
GDTI	0.09	0.12	1.60	2.21	0~0.57	135.45
GSI	0.08	0.11	1.43	1.44	0~0.48	129.00
MFV	0.15	0.19	1.41	1.60	0~1	121.81

RGR, relative germination rate; RGE, relative germination energy; GDTI, germination drought tolerant index; GSI, germination stress index; MFV, membership function value; SD, standard deviation; CV, coefficient of variation.

**Table 3 biology-11-01812-t003:** Analysis of variance of four drought tolerance indices of 410 accessions at 0% PEG and 20% PEG.

Trait	Source	DF	Sum of Square	Mean Square	*F* Value	Pr > F
RGR	Geno	409	44.94	0.11	11.67	<0.0001
	Block	2	0.01	0.00	0.28	0.75
RGE	Geno	409	39.22	0.10	10.94	<0.0001
	Block	2	0.01	0.00	0.47	0.63
GDTI	Geno	409	13.88	0.03	9.01	<0.0001
	Block	2	0.00	0.00	0.28	0.75
GSI	Geno	409	12.41	0.03	8.19	<0.0001
	Block	2	0.00	0.00	0.16	0.86

RGR, relative germination rate; RGE, relative germination energy; GDTI, germination drought tolerant index; GSI, germination stress index; DF, degree of freedom.

**Table 4 biology-11-01812-t004:** Phenotypic correlations between five drought tolerance indices in the 410 soybean accessions.

Trait	RGR	RGE	GDTI	GSI
RGE	0.9840 ***			
GDTI	0.9640 ***	0.9766 ***		
GSI	0.9490 ***	0.9524 ***	0.9729 ***	
MFV	0.9862 ***	0.9902 ***	0.9909 ***	0.9819 ***

*** significant level under 0.0001 for Pearson correlation test. RGR, relative germination rate; RGE, relative germination energy; GDTI, germination drought tolerant index; GSI, germination stress index; MFV, Membership function value.

**Table 5 biology-11-01812-t005:** SNPs significantly associated with five drought tolerance indices (−log*p* > 4.5).

Marker	Chr	Position	Associated Traits (R^2^)	Reported QTLs/Genes
Gm01_35877607	1	35877607	RGR(7.20), RGE(7.53), GDTI(6.93), MFV(6.95)	Seed set [5]; seed weight [40,41]
Gm01_38948188	1	38948188	GDTI(6.73)	Pod wall weight [42]
Gm01_47042336	1	47042336	GSI(6.31), GDTI(6.51)	Drought index [1]; root area [43]; root length [43]
Gm01_48619013	1	48619013	GDTI(6.16)	Drought index [1]
Gm02_6357585	2	6357585	GDTI(7.11)	Canopy wilt [7]
Gm03_39037	3	39037	GDTI(6.08)	
Gm04_4484515	4	4484515	RGR(6.84), RGE(7.22), GDTI(6.99), MFV(6.74)	Canopy wilt [7]; seed set [5]; seed weight [44]
Gm04_50945875	4	50945875	GDTI(6.69)	Seed number [44]; WUE [23]
Gm05_38540838	5	38540838	RGR(7.60)	Cellwall polysacch composition [45]
Gm06_9791913	6	9791913	GDTI(6.17)	Seed weight [46]; shoot weight [47]
Gm07_24735482	7	24735482	GDTI(6.59)	
Gm08_1438457	8	1438457	RGE(6.08)	Seed weight per plant [48]
Gm08_4052111	8	4052111	GDTI(6.91)	Canopy wilt [7]; seed weight [49]
Gm08_7972856	8	7972856	RGR(8.03)	Root density, lateral [50]; seed set [5]
Gm09_11414508	9	11414508	RGE(5.19), GDTI(6.00), MFV(5.32)	Seed yield [51,52]
Gm09_18023730	9	18023730	GSI(6.46), GDTI(6.77), MFV(6.03)	Seed yield [52,53]
Gm11_30280479	11	30280479	RGR(8.33), RGE(7.04), GDTI(8.08), MFV(7.53)	Seed set [5]
Gm13_35517964	13	35517964	GDTI(6.55)	
Gm14_46603856	14	46603856	GDTI(6.19)	Canopy wilt [20]
Gm15_11950665	15	11950665	GDTI(6.52)	Seed weight [48]
Gm15_47429024	15	47429024	GSI(6.45)	
Gm19_49449499	19	49449499	GDTI(6.46)	Canopy wilt [7]; drought tolerance [13]
Gm20_4618170	20	4618170	GDTI(6.28)	
Gm20_13921498	20	13921498	RGR(6.66), RGE(7.26), GDTI(7.01), MFV(6.64)	Seed weight [41]
Gm20_34956219	20	34956219	RGR(6.88), RGE(7.87), GSI(7.22), GDTI(8.33), MFV(7.81)	Canopy wilt [20]; root density, lateral [54]; seed set [41]; WUE [24]
Gm20_36902659	20	36902659	RGR(7.69), RGE(8.03), GSI(8.19), GDTI(9.66), MFV(8.57)	Root density, lateral [54]

RGR, relative germination rate; RGE, relative germination energy; GDTI, germination drought tolerant index; GSI, germination stress index; MFV, Membership function value.

## Data Availability

Not applicable.

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
