# Peer review of "Identification of Drought-Tolerance Genes in the Germination Stage of Soybean"

_biology, 2022, doi:10.3390/biology11121812_

Round 1

Reviewer 1 Report

Reading your work, I am full of admiration for the enormous amount of research you have done while analyzing as many as 410 different genotypes of soybeans. You identified 26 different SNPs related to drought tolerance of plants. Your research shows how important drought is, which is starting to cause more and more problems in different regions of the world, and that by analyzing and learning about drought tolerance genes, it will be possible in the future to select and cultivate soybean to increase its resistance to drought. The work is written correctly and contains all the components that a scientific work should contain. The selection of references was correct, the applied research methods were correctly selected and the results were subjected to statistical analysis. I believe that the work deserves publication, therefore I will recommend the Editorial Board to accept it for publication.

Author Response

Dear Editor,

We greatly appreciate the critical comments from you and the reviewers for our manuscript submitted to Biology (ID: biology-1937485). We are very sorry that there were unclear explanations shown in previous version. We have revised the manuscript thoroughly based on your comments. The detailed response to each comment was attached below.

We would like to resubmit our manuscript to Biology for the consideration of publication. Thank you again for reviewing our manuscript, and please let us know if you have any additional comments.

Sincerely yours,

Lijuan Qiu and coauthors

Comments from Reviewer : Reading your work, I am full of admiration for the enormous amount of research you have done while analyzing as many as 410 different genotypes of soybeans. You identified 26 different SNPs related to drought tolerance of plants. Your research shows how important drought is, which is starting to cause more and more problems in different regions of the world, and that by analyzing and learning about drought tolerance genes, it will be possible in the future to select and cultivate soybean to increase its resistance to drought. The work is written correctly and contains all the components that a scientific work should contain. The selection of references was correct, the applied research methods were correctly selected and the results were subjected to statistical analysis. I believe that the work deserves publication, therefore I will recommend the Editorial Board to accept it for publication.

Answer: We appreciated your comments. We hope to get more support and help from you.

Reviewer 2 Report

The manuscript presented by Zhao et al. “Identification of drought-tolerance genes in the germination stage of soybean” is identified the drought-tolerance genes.

The figures and tables are missing in the manuscript.

Write the objective of the study in the Introduction section.

Data presented in the manuscript are less and need to add more related data.

Finally, the discussion section not clearly described the results.

Based on the above considerations I cannot recommend the publication of this research paper in its current form.

Author Response

Dear Editor,

We greatly appreciate the critical comments from you and the reviewers for our manuscript submitted to Biology (ID: biology-1937485). We are very sorry that there were unclear explanations shown in previous version. We have revised the manuscript thoroughly based on your comments. The detailed response to each comment was attached below.

We would like to resubmit our manuscript to biology for the consideration of publication. Thank you again for reviewing our manuscript, and please let us know if you have any additional comments.

Sincerely yours,

Lijuan Qiu and coauthors

Comments from Reviewer #2: The manuscript presented by Zhao et al. “Identification of drought-tolerance genes in the germination stage of soybean” is identified the drought-tolerance genes. The figures and tables are missing in the manuscript.

Answer: We are sorry for the carelessness. We have added tables and pictures in the revised manuscript.

Comments from Reviewer #2: Write the objective of the study in the Introduction section.

Answer: We appreciated your comments. The objective of the study was added in the introduction section accordingly.

Comments from Reviewer #2: Data presented in the manuscript are less and need to add more related data.

Answer: We are sorry for the carelessness. We present the main research results in the form of tables and figures in the manuscript, upload the phenotype data in the form of attachments to the submission website, and upload the genotype data to the public database.

Comments from Reviewer #2: Finally, the discussion section not clearly described the results.

Answer: Sorry for ambiguous points in the manuscript. The manuscript has been revised and checked throughout by native speaker with experience of scientific writing.

Reviewer 3 Report

The manuscript is clear, relevant for the field and present in a well-structured manner.

The tables and figures are not visible! Missing?

          Kindly recommend to continue studying drought-tolerance genes and markers to assist with the selection and development of drought-tolerant soybean accessions.

Author Response

We appreciated your comments. We have added tables and pictures in revised manuscript, and We will continue to study drought tolerance in soybean with the hope of discovering related genes and developing molecular markers for application in soybean variety improvement.

Round 2

Reviewer 3 Report

The suggested changes have been made in the new attached manuscript. I really appreciate it. Thank you! The tables data was adequate to test the proposed hypotheses and the figures are clear and easy to interpret and understand.
